# SyMoFlow: Interaction-Aware Motion Synthesis from Text via Symmetric Flows

## Abstract

Human-Human Interaction (HHI) generation aims to synthesize plausible and co-ordinated motion sequences for multiple agents in a shared environment. Existing approaches often struggle to capture reciprocal dependencies, maintain semantic alignment with textual descriptions, or balance realism and diversity. To address these challenges, we propose SyMoFlow, a text-driven motion synthesis framework that leverages an interaction-symmetric decomposition of the joint motion distribution. SyMoFlow generates sequential single-agent motions: it first produces an interaction-aware motion for one agent conditioned on text, then synthesizes the second agent's motion conditioned on the first, capturing both prior action and reciprocal reaction. By explicitly modeling interdependent dynamics, our approach produces coordinated, causally consistent behaviors while allowing flexible flow-based sampling to enhance multimodality and diversity. Extensive experiments on the InterHuman and InterX benchmarks demonstrate that SyMoFlow achieves state-of-the-art realism and text alignment while significantly improving the diversity of plausible interactions.

## 1 Introduction

Human-Human Interaction (HHI) generation aims to synthesize plausible and coordinated motion sequences for multiple agents operating within a shared environment. Unlike single-person motion generation, which models individual action dynamics independently (Tevet et al., 2023; Guo et al., 2022a; Shafir et al., 2024), HHI generation must capture the complex coordination and reciprocal dependencies between interacting agents. Each agent's behavior is often influenced by subtle cues from the other, making it challenging to produce sequences that are both realistic and temporally coherent (Liang et al., 2024; Javed et al., 2025; Wang et al., 2025b).

Some recent methods, such as InterGen (Liang et al., 2024) and InterMask (Javed et al., 2025), attempt to model interacting agents via a joint distribution over both actors' motions to capture dependencies directly. While this joint modeling can improve coordination, these approaches often struggle to accurately align multiple plausible motion modes with the textual description and face a trade-off between motion realism and diversity. Existing approaches like FreeMotion (Fan et al., 2024) typically decompose the task into single-agent motion generation followed by a reaction module. Although this decomposition simplifies modeling, it treats the initial single-agent motion independently of the interacting partner, which may result in incoherent, delayed, or contextually inconsistent responses. Consequently, despite progress in realism, diversity, and semantic alignment, current methods still face challenges in jointly ensuring interaction consistency, capturing causal dependencies, and generating coordinated multi-agent behaviors.

To address these challenges, we propose **SyMoFlow**, an interaction-aware, text-driven motion synthesis framework. SyMoFlow models the joint dynamics of two interacting agents through an *interaction-symmetric* decomposition, where interaction symmetry reflects the reciprocal nature of human interactions: for any action performed by one agent, there exists a plausible and contextually consistent response from the other. By explicitly modeling these dependencies, SyMoFlow generates coordinated multi-agent motions that achieve realism comparable to state-of-the-art methods while significantly enhancing the diversity of plausible reactions and maintaining strong alignment with textual prompts.

Specifically, SyMoFlow first generates an interaction-aware motion for one agent conditioned on the textual description, and then synthesizes the second agent's motion conditioned on the first, capturing both prior action and reciprocal reaction in a unified sequential framework. This sequential design reduces the high-dimensional multi-agent generation problem to a tractable series of single-agent generations while explicitly modeling interdependent dynamics between agents. The generative process is implemented via a discrete flow matching path that interpolates between an initial latent noise and the target motion tokens, allowing flexible intermediate sampling strategies, controllable stochasticity, and smooth transitions. This flow path design not only preserves realism and semantic alignment with textual prompts, but also substantially improves the diversity and multimodality of plausible reactions across generated sequences.

In summary, our contributions are threefold: (1) we introduce SyMoFlow, an interaction-symmetric sequential generation framework for text-driven, interaction-aware motion synthesis that explicitly models reciprocal dependencies between agents; (2) we highlight a flexible flow-based generative path that enables diverse, multimodal reaction sampling while maintaining realism; and (3) we demonstrate that SyMoFlow achieves a favorable balance of realism, diversity, and text alignment, producing coordinated, plausible, and varied HHI sequences on the InterHuman and InterX benchmarks (Liang et al., 2024; Xu et al., 2024a).

## 2 RELATED WORK

### 2.1 HUMAN MOTION SYNTHESIS

Human motion generation aims to construct realistic and diverse 3D motions from multiple input modalities. Single-person motion generation has been widely studied, focusing on text (Tevet et al., 2023; Petrovich et al., 2022; 2023), music (Li et al., 2020; Jiang et al., 2023), semantic action labels (Guo et al., 2020; Petrovich et al., 2021; Xu et al., 2023), or physics-based constraints (Yuan et al., 2023). Generative techniques include GANs (Xu et al., 2023), VAEs (Guo et al., 2022b; Jiang et al., 2023), diffusion models (Shafir et al., 2024; Tevet et al., 2023; Yuan et al., 2023), and flow-based models (Aliakbarian et al., 2022). Another line of research leverages discrete representations via VQ-VAE (Zhang et al.; Wang et al., 2024; Lee et al., 2024), enabling autoregressive token prediction and downstream pre-trained models to enhance semantic understanding and diversity. Despite these advances, single-person methods do not address the dynamics of multi-agent interactions.

### 2.2 MULTI-PERSON MOTION AND REACTION GENERATION

Multi-person motion generation, especially for close Human-Human Interactions (HHI), is more challenging due to dependencies between agents. Existing work advances along two main directions: (i) jointly modeling both agents' motions via shared latent spaces or diffusion processes; (ii) explicitly capturing actor-reactor interaction dynamics with attention or temporal correlation modules. Representative methods include InterGen (Liang et al., 2024), in2IN (Ponce et al., 2024), FreeMotion (Fan et al., 2024), InterMask (Javed et al., 2025), and TIMotion (Wang et al., 2025b).

Existing approaches for interaction generation can be categorized into three groups. The first group models the joint distribution of interacting agents (Liang et al., 2024; Javed et al., 2025), improving realism, diversity, or semantic consistency. However, joint modeling is inflexible and can only capture interaction relationships implicitly. The second group focuses primarily on reaction generation (Chopin et al., 2023; Xu et al., 2024b), producing a response motion conditioned on a given agent; however, these methods cannot be directly applied to full two-person interactions. The third group employs distribution decomposition, such as FreeMotion (Fan et al., 2024), which decomposes motion into a primary actor and a conditioned reaction; while flexible, such approaches often reduce inter-agent coordination and consistency due to separate modeling. In contrast, SyMoFlow adopts an interaction-symmetric formulation that jointly models both the prior and posterior motion distributions of actor and reactor, achieving coordinated, causally consistent, and realistic interactions in a single-stage training process without additional supervision.

## 2.3 FLOW MATCHING FOR MOTION GENERATION

Flow Matching (Lipman et al., 2022) has recently emerged as a promising alternative to diffusion-based generative models. By constructing linear trajectories between source and target distributions and solving ordinary differential equations (ODEs), flow matching simplifies training objectives and accelerates inference compared to diffusion. It has achieved competitive or superior results across multiple modalities, including images (Dao et al., 2023), audio (Guan et al., 2024), and video (Jin et al., 2024). In motion generation, MotionFlow (Meral et al., 2024) demonstrates that flow-based models can match the quality of diffusion-based baselines while enabling faster sampling. The ability of flow matching to model transitions between arbitrary distributions makes it particularly well-suited for paired data scenarios, such as capturing the dynamics of actor–reactor motions.

However, existing applications of flow matching in human motion synthesis primarily focus on single-agent trajectories or static pose distributions, leaving dynamic multi-agent interactions largely unexplored. Recent advances in Discrete Flow Matching (DFM) (Gat et al., 2024a; Wang et al., 2025a) extend the framework from continuous spaces to discrete latent variables, broadening its applicability to structured data such as language and quantized motion tokens. Building on these developments, SyMoFlow leverages DFM to jointly model the causal and conditional dependencies between interacting agents. This enables high-fidelity generation while promoting motion diversity, semantic alignment with textual prompts, and coordination between actor and reactor. In doing so, it addresses the limitations of previous multi-person motion synthesis methods and highlights the potential of flow-based models in interaction-aware scenarios.

## 3 METHODOLOGY

We present **SyMoFlow**, a sequential framework for human-human interaction generation that explicitly models *interaction-symmetric* dynamics. SyMoFlow captures the joint behavior of two interacting agents through sequential interaction-aware single-motion generation. Specifically, it first generates a plausible motion for one agent and then conditions on this motion to synthesize the other agent's response, effectively modeling both prior actions and reciprocal reactions in a unified framework. By exploiting interaction symmetry between agents, SyMoFlow supports parameter sharing and one-stage training, while ensuring that the generated sequences are coherent, coordinated, and causally consistent.

### 3.1 PROBLEM FORMULATION

Let $\mathbf{x}^A = (\mathbf{x}_1^A, \ldots, \mathbf{x}_L^A)$ and $\mathbf{x}^B = (\mathbf{x}_1^B, \ldots, \mathbf{x}_L^B)$ denote the motion sequences of two interacting agents. SyMoFlow factorizes the joint distribution into sequential single-motion generations:

$$p(\mathbf{x}^A, \mathbf{x}^B) = p_\theta(\mathbf{x}^A)\, p_\phi(\mathbf{x}^B \mid \mathbf{x}^A), \tag{1}$$

where $p_\theta(\mathbf{x}^A)$ models the prior motion of the first agent, and $p_\phi(\mathbf{x}^B \mid \mathbf{x}^A)$ generates the second agent's motion conditioned on the first.

We further impose a *interaction-symmetric property* between the two agents:

$$p_\phi(\mathbf{x}^B \mid \mathbf{x}^A) = p_\phi(\mathbf{x}^A \mid \mathbf{x}^B), \tag{2}$$

which reflects the reciprocal nature of human-human interactions. This symmetry allows either agent to serve as the initiator, avoids biasing the model toward a fixed action-reaction order, and facilitates shared parameters for efficient, one-stage training while preserving interaction coherence.

### 3.2 FRAMEWORK

Figure 1 provides an overview of the SyMoFlow motion generation pipeline. Continuous motion sequences are first encoded into discrete tokens using a VQ-VAE, which converts high-dimensional joint trajectories into a compact symbolic space. Interaction-aware motion generation is then formulated as a discrete flow matching (DFM) process, where a transformer-based architecture predicts probability velocity fields that guide the flow from latent noise to target motion tokens. In this design, the VQ-VAE provides the discrete motion representation, DFM defines the generative framework, and the transformer models complex spatio-temporal dependencies within the flow, enabling coordinated, realistic, and diverse interactions.

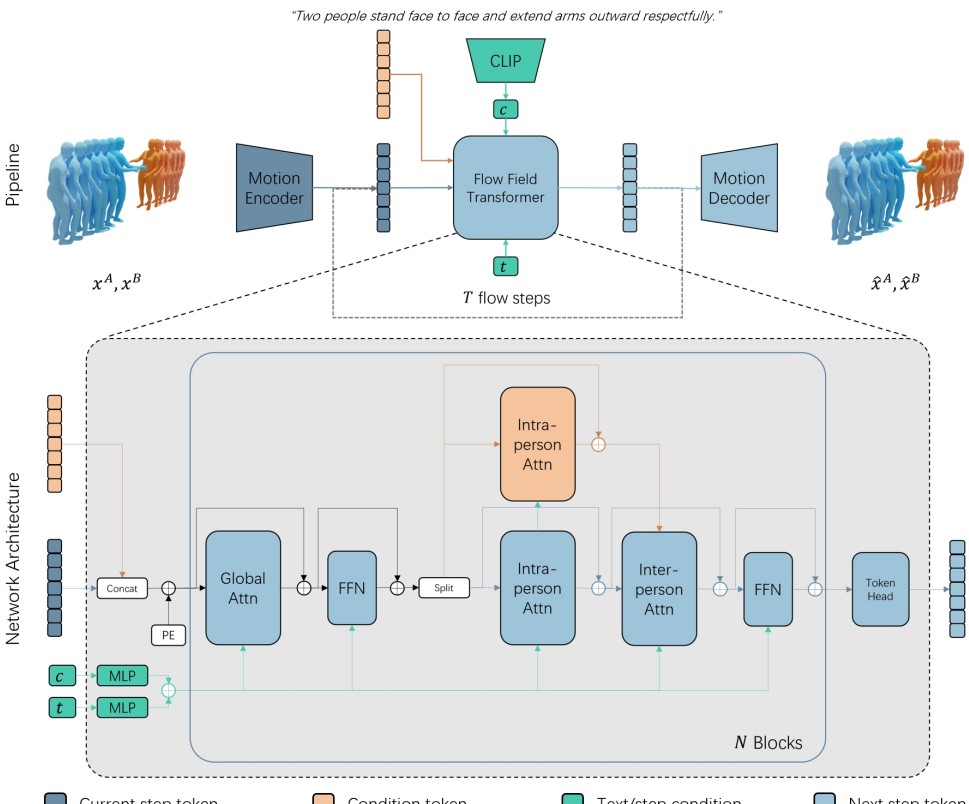

Figure 1: Overview of the SyMoFlow architecture. Motion sequences are first discretized by a shared VQ-VAE, then modeled with transformer blocks that integrate global, intra-person, and inter-person attentions. During training, discrete flow matching leverages interaction-symmetric paths for motion and reaction consistency, while during inference, an interaction-aware sequential process generates the first agent's motion from noise and conditions the second agent's reaction on it.

### 3.2.1 VQ-VAE Discretization of Motion Sequences

To enable discrete flow-based generation, each individual motion sequence $\mathbf{x}^A, \mathbf{x}^B \in \mathbb{R}^{T \times J \times d}$, with $T$ frames, $J$ joints, and $d$-dimensional features, is encoded via a 2D convolutional encoder, following InterMask (Javed et al., 2025). Temporal and spatial dimensions are downsampled to a latent tensor $\tilde{\mathbf{z}} \in \mathbb{R}^{n \times j \times d'}$ with $n \ll T$ and $j \ll J$. Each latent vector is quantized to the nearest codebook entry $\mathcal{C} = \{e_k\}_{k=1}^{|\mathcal{C}|} \subset \mathbb{R}^{d'}$, producing discrete tokens $\mathbf{z} \in \{0, 1, \ldots, |\mathcal{C}| - 1\}^{n \times j}$. A shared decoder reconstructs motions from these tokens, ensuring both agents share the same discrete latent space.

This discretization provides a compact symbolic representation that forms the basis for DFM-based motion synthesis. For full implementation and training details, including VQ-VAE loss functions and geometric regularization, see Appendix A.

### 3.2.2 Discrete Flow Matching for Interaction Generation

Interaction-aware motion synthesis is realized via *Discrete Flow Matching* (DFM) (Gat et al., 2024b), which models the transition from latent noise to discrete motion tokens. Each agent's motion tokens are updated sequentially along a continuous probability path, conditioned on the other agent's motion. This sequential refinement allows the model to maintain semantic alignment with textual prompts, capture reciprocal dependencies between agents, and generate diverse plausible reactions.

Concretely, motion sequences are first encoded into discrete tokens via a VQ-VAE codebook. DFM defines a family of intermediate distributions $\{p_t(z)\}_{t \in [0,1]}$ interpolating between the noise prior

$p(z)$ and the empirical data distribution $q(z)$:

$$p_t(z) = (1 - \kappa_t)p(z) + \kappa_t q(z), \quad \kappa_0 = 0, \ \kappa_1 = 1, \tag{3}$$

where $\kappa_t$ is a time-dependent scheduler. This construction allows tokens to gradually transition from noise to data-consistent states.

To simulate this evolution, DFM introduces a probability velocity field $u_t$ that governs token transitions in a continuous-time Markov chain (CTMC), ensuring consistency with the path:

$$\frac{d}{dt}p_t(z) + \text{div}_z\big(p_t u_t\big) = 0. \tag{4}$$

Sampling thus amounts to evolving token trajectories along $t \in [0, 1]$ under these dynamics, progressively replacing noise with structured motion tokens.

In our framework, each agent's motion tokens are obtained by encoding and discretizing the corresponding continuous motion sequences via the shared VQ-VAE: $\mathbf{x}^A \to \mathbf{z}^A$ and $\mathbf{x}^B \to \mathbf{z}^B$. During generation, the first agent's tokens $\mathbf{z}^A$ are sampled from a uniform noise prior $p(\mathbf{z}^A)$, which serves as the source distribution for producing the target latent sequence distribution $q(\mathbf{z}^A)$. The second agent's tokens $\mathbf{z}^B$ are then generated conditionally on the first agent, using $p(\mathbf{z}^B) = q(\mathbf{z}^A)$ as the source distribution to generate $q(\mathbf{z}^B)$. This interaction-symmetric factorization explicitly captures reciprocal action–reaction dependencies, ensuring that the generated sequences are coordinated and causally consistent (see Figure 4 in Appendix). Further formal definitions of the discrete probability paths, and the full CTMC-based sampling procedure are provided in Appendix B.

### 3.2.3 TRANSFORMER BLOCK FOR SPATIO-TEMPORAL MOTION MODELING

To capture complex spatio-temporal interactions in two-person motions, we employ a transformer block architecture inspired by InterGen (Liang et al., 2024) and InterMask (Javed et al., 2025) as shown in Figure 1. Each block integrates global self-attention to model long-range dependencies, intra-person spatio-temporal attention for individual motion patterns, and inter-person cross-attention to capture interactions between agents. The block is conditioned on the textual prompt $c$, the current DFM time step, and an auxiliary motion condition $\mathbf{z}_{\text{cond}}$, with conditioning incorporated into global self-attention via concatenation and into inter-person cross-attention through keys and values. Text and time-step conditioning are applied via adaptive layer normalization (Peebles & Xie, 2023). Full details on attention mechanisms, token slicing, and conditioning strategy are provided in Appendix C.

### 3.3 TRAINING

The training objective of SyMoFlow is designed to produce interaction-aware, physically plausible, and semantically aligned motion sequences. The model learns to predict the target motion tokens from noised inputs along the discrete flow matching (DFM) path while respecting geometric consistency and inter-agent dependencies. In essence, the training combines a sequence-level prediction loss with motion-level consistency constraints, encouraging both coordination and realism.

**DFM Cross-Entropy Loss.** The discrete flow matching model is trained to generate coherent interactions by predicting the ground-truth latent sequence $\mathbf{z}_1 \sim q(\cdot)$ from a noised input $\mathbf{z}_t$, conditioned on the textual prompt $c$ and an auxiliary motion condition $\mathbf{z}_{\text{cond}}$. The auxiliary condition can be either the other agent's motion codes (e.g., $\mathbf{z}^A$ or $\mathbf{z}^B$) or random noise $\epsilon \sim \mathcal{U}(0, 1)$, with half of the sequences replaced with noise to improve robustness.

At each training step, a continuous time $t \in [0, 1]$ is uniformly sampled, and $\mathbf{z}_t$ is generated along the DFM probability path $p_t(\cdot \mid \mathbf{z}_1)$, which interpolates between the target sequence and the condition according to the scheduler $\kappa_t$. The model receives $\mathbf{z}_t$, the text $c$, and $\mathbf{z}_{\text{cond}}$, and predicts per-token logits for $\mathbf{z}_1$. The cross-entropy loss is defined as:

$$\mathcal{L}_{\text{CE}} = \mathbb{E}_{t \sim \mathcal{U}[0,1], \ \mathbf{z}_1 \sim q(\cdot), \ \mathbf{z}_t \sim p_t(\cdot|\mathbf{z}_1)} \left[ -\sum_{i=1}^{L} \log p_\theta(z_1^i \mid \mathbf{z}_t, c, \mathbf{z}_{\text{cond}}) \right]. \tag{5}$$

**Motion Consistency Loss.** To enforce physically plausible motions, the predicted logits are converted to probabilities via softmax and decoded into continuous motion sequences:

$$\hat{\mathbf{x}}_t = \text{Decode}\big(f_\theta(\mathbf{z}_t, c, \mathbf{z}_{\text{cond}})\big), \tag{6}$$

where $f_\theta(\cdot)$ outputs the per-token logits. The motion consistency loss penalizes deviations in velocity, foot contact, and bone length:

$$\mathcal{L}_{\text{consis}} = \mathbb{E}_{\mathbf{x}_t}\Big[\lambda_{\text{vel}}\,\mathcal{L}_{\text{vel}}(\mathbf{x}, \hat{\mathbf{x}}_t) + \lambda_{\text{fc}}\,\mathcal{L}_{\text{fc}}(\mathbf{x}, \hat{\mathbf{x}}_t) + \lambda_{\text{bl}}\,\mathcal{L}_{\text{bl}}(\mathbf{x}, \hat{\mathbf{x}}_t)\Big], \tag{7}$$

where $\mathbf{x}$ is the ground-truth motion corresponding to $\mathbf{z}_t$, and $\lambda_{\text{vel}}, \lambda_{\text{fc}}, \lambda_{\text{bl}}$ control the relative weight of each geometric term. During training, $\mathbf{z}_{\text{cond}}$ alternates between the other agent's motion codes and noise; during inference, it is set to the previously generated motion (e.g., $\mathbf{z}^A$ when generating $\mathbf{z}^B$) to ensure coherent interactions.

**Overall Objective.** The final training loss combines sequence prediction and motion consistency:

$$\mathcal{L} = \mathcal{L}_{\text{CE}} + \mathcal{L}_{\text{consis}}. \tag{8}$$

### 3.4 INFERENCE

During inference, interaction sequences are generated in a two-stage process, following the discrete flow matching framework (see Appendix B.2 for details):

**Interaction-aware single motion generation (agent A):** We initialize the latent sequence $\mathbf{z}_0$ and the auxiliary condition $\mathbf{z}_{\text{cond}}$ from independent uniform noise:

$$\mathbf{z}_0 \sim \mathcal{U}(0, 1), \quad \mathbf{z}_{\text{cond}} \sim \mathcal{U}(0, 1). \tag{9}$$

conditioned on the textual description $c$, the model then evolves $\mathbf{z}_0$ along the DFM path to obtain the motion latent sequence $\mathbf{z}^A$.

**Reaction generation (agent B):** We initialize both the latent sequence and auxiliary condition with the previously generated motion of agent A:

$$\mathbf{z}_0 = \mathbf{z}^A, \quad \mathbf{z}_{\text{cond}} = \mathbf{z}^A, \tag{10}$$

along with the same text condition $c$. The model then generates the reaction sequence $\mathbf{z}^B$ along the corresponding DFM path.

At each step, the model outputs per-token logits conditioned on the current latent sequence, the text condition $c$, and $\mathbf{z}_{\text{cond}}$, which can be decoded into motions if motion consistency evaluation is applied. This procedure ensures that agent B's motions are coherent with both the text prompt and agent A's generated motions.

## 4 EXPERIMENTAL RESULTS

### 4.1 DATASETS

We conduct experiments on two benchmark datasets for text-conditioned human-human interaction generation: InterHuman (Liang et al., 2024) and InterX (Xu et al., 2024a). InterHuman contains 7,779 interaction sequences, while InterX contains 11,388 sequences, each paired with three distinct textual annotations. These datasets cover diverse interaction scenarios and serve as standard testbeds for evaluating HHI generation models. For implementation and representation details, please refer to Appendix D.

### 4.2 EVALUATION PROTOCOL

We evaluate SyMoFlow on both standard Human-Human Interaction (HHI) generation and reaction generation tasks. In the HHI task, the model generates coordinated two-person motion sequences conditioned on a textual description. For reaction generation, the model produces the motion of a second agent given both the textual prompt and the first agent's motion, simulating a responsive interaction scenario.

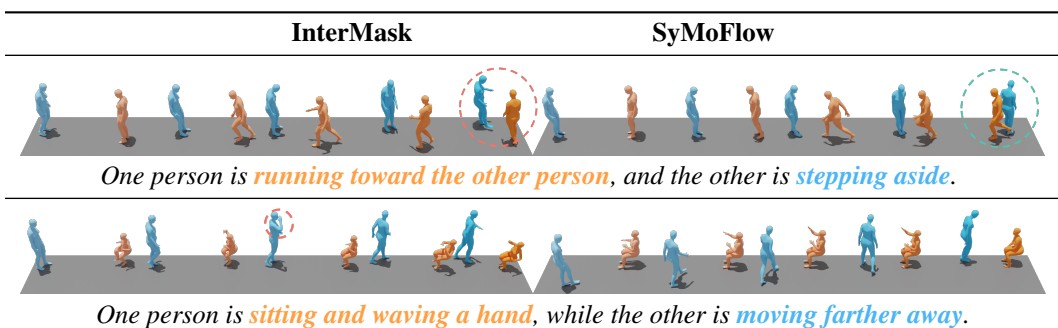

| InterMask | SyMoFlow |

*One person is *running toward the other person*, and the other is *stepping aside*.*

*One person is *sitting and waving a hand*, while the other is *moving farther away*.*

Figure 2: Qualitative comparison between SyMoFlow and InterMask (Javed et al., 2025).

**Metrics.** Following prior work (Liang et al., 2024), we adopt feature-space evaluation metrics to comprehensively assess model performance. Frechet Inception Distance (FID) measures the fidelity of generated motions by comparing their feature distribution with that of real interactions. R-Precision quantifies semantic alignment, evaluating how well generated sequences correspond to their textual prompts. MM-Dist assesses realism and alignment by measuring distances in a learned motion feature space. Diversity captures the overall variation among generated sequences for the same textual input, while Multimodality (MModality) evaluates the model's ability to produce multiple distinct and plausible interactions conditioned on the same text.

Our evaluation protocol directly compares SyMoFlow against state-of-the-art methods for both HHI and reaction generation tasks, demonstrating its effectiveness in fidelity, text alignment, diversity, and multimodality.

### 4.3 HUMAN-HUMAN INTERACTION GENERATION RESULTS

**Quantitative Comparison.** SyMoFlow demonstrates competitive performance to existing methods on both the InterHuman and InterX datasets. As shown in Table 1, it consistently outperforms representative interaction-aware baselines, including InterGen (Liang et al., 2024), in2IN (Ponce et al., 2024), FreeMotion (Fan et al., 2024), InterMask (Javed et al., 2025), and TIMotion (Wang et al., 2025b). On InterHuman, SyMoFlow attains the second-lowest FID while leading in R-Precision and MM-Dist, demonstrating superior motion realism, semantic alignment with textual prompts, and accurate modeling of inter-agent dependencies. For a fair comparison, we report the results of TIMotion with its transformer backbone. Compared to InterMask and other baselines, SyMoFlow achieves a better balance between diversity and multimodality, capturing a wider range of plausible reactions without sacrificing interaction plausibility.

**Qualitative Comparison.** Visual inspection further confirms that SyMoFlow produces temporally coherent and interaction-aware motions, correctly anticipating the first agent's actions and generating contextually appropriate responses for the second agent. The outputs exhibit more expressive and semantically consistent behaviors, particularly in scenarios involving close interactions or coordinated movements. Representative visualizations are shown in Figure 5, with additional comparisons against baselines in Figure 2 and more qualitative examples in Figure 6. These comparisons highlight SyMoFlow's ability to synthesize diverse, realistic, and contextually aligned two-person interactions.

### 4.4 REACTION GENERATION RESULTS

**Quantitative Comparison.** We evaluate SyMoFlow on the task of interaction-aware reaction generation using the InterHuman dataset. Table 2 summarizes the quantitative results. Our method outperforms prior approaches, including InterMask, across most metrics. Specifically, SyMoFlow achieves higher R-Precision, indicating better semantic alignment with the textual prompt, while also attaining lower FID in reaction generation, reflecting more realistic motion outputs. Diversity and multimodality scores further demonstrate the model's ability to generate varied and plausible reactions that respect the interaction context.

| Dataset | Method | R-Precision↑ | | | FID↓ | MMDist↓ | Diversity→ | MModality↑ |
| --- | --- | --- | --- | --- | --- | --- | --- | --- |
| | | Top-1 | Top-2 | Top-3 | | | | |
| | Ground Truth | $0.452^{\pm0.008}$ | $0.610^{\pm0.009}$ | $0.701^{\pm0.008}$ | $0.273^{\pm0.007}$ | $3.755^{\pm0.008}$ | $7.948^{\pm0.064}$ | – |
| | T2M (Guo et al., 2022a) | $0.153^{\pm0.012}$ | $0.260^{\pm0.009}$ | $0.339^{\pm0.012}$ | $9.167^{\pm0.056}$ | $7.125^{\pm0.018}$ | $7.602^{\pm0.045}$ | $1.387^{\pm0.076}$ |
| | MDM (Tevet et al., 2023) | $0.223^{\pm0.009}$ | $0.334^{\pm0.008}$ | $0.466^{\pm0.010}$ | $7.069^{\pm0.054}$ | $6.212^{\pm0.021}$ | $7.244^{\pm0.038}$ | $\underline{2.350}^{\pm0.080}$ |
| | ComMDM (Shafir et al., 2024) | $0.371^{\pm0.010}$ | $0.515^{\pm0.012}$ | $0.624^{\pm0.010}$ | $5.918^{\pm0.079}$ | $5.108^{\pm0.014}$ | $7.387^{\pm0.029}$ | $1.822^{\pm0.052}$ |
| InterHuman | InterGen (Liang et al., 2024) | $0.449^{\pm0.004}$ | $0.591^{\pm0.003}$ | $0.666^{\pm0.004}$ | $5.674^{\pm0.085}$ | $\underline{3.790}^{\pm0.001}$ | $8.021^{\pm0.035}$ | $2.141^{\pm0.063}$ |
| | in2IN (Ponce et al., 2024) | $0.425^{\pm0.008}$ | $0.576^{\pm0.008}$ | $0.662^{\pm0.009}$ | $5.535^{\pm0.120}$ | $3.803^{\pm0.002}$ | $7.953^{\pm0.047}$ | $1.295^{\pm0.023}$ |
| | FreeMotion (Fan et al., 2024) | $0.326^{\pm0.003}$ | $0.462^{\pm0.006}$ | $0.544^{\pm0.006}$ | $6.740^{\pm0.130}$ | $3.848^{\pm0.002}$ | $7.828^{\pm0.130}$ | $1.226^{\pm0.046}$ |
| | InterMask (Javed et al., 2025) | $0.449^{\pm0.004}$ | $0.599^{\pm0.005}$ | $0.683^{\pm0.004}$ | $\mathbf{5.154}^{\pm0.061}$ | $\underline{3.790}^{\pm0.002}$ | $\mathbf{7.944}^{\pm0.033}$ | $1.737^{\pm0.020}$ |
| | TIMotion (Wang et al., 2025b) | $\mathbf{0.491}^{\pm0.005}$ | $\underline{0.648}^{\pm0.004}$ | $\underline{0.724}^{\pm0.004}$ | $5.433^{\pm0.080}$ | $\mathbf{3.775}^{\pm0.001}$ | $8.032^{\pm0.030}$ | $0.952^{\pm0.032}$ |
| | Ours | $\mathbf{0.491}^{\pm0.006}$ | $\mathbf{0.653}^{\pm0.005}$ | $\mathbf{0.730}^{\pm0.005}$ | $\underline{5.416}^{\pm0.078}$ | $\mathbf{3.775}^{\pm0.001}$ | $\underline{7.953}^{\pm0.037}$ | $\mathbf{2.844}^{\pm0.095}$ |
| | Ground Truth | $0.429^{\pm0.004}$ | $0.626^{\pm0.003}$ | $0.736^{\pm0.003}$ | $0.002^{\pm0.0002}$ | $3.536^{\pm0.013}$ | $9.734^{\pm0.078}$ | – |
| | T2M (Guo et al., 2022a) | $0.184^{\pm0.010}$ | $0.298^{\pm0.006}$ | $0.396^{\pm0.005}$ | $9.576^{\pm0.006}$ | $9.576^{\pm0.006}$ | $2.771^{\pm0.151}$ | $2.761^{\pm0.042}$ |
| | MDM (Tevet et al., 2023) | $0.203^{\pm0.009}$ | $0.329^{\pm0.007}$ | $0.426^{\pm0.005}$ | $23.701^{\pm0.057}$ | $9.548^{\pm0.014}$ | $5.856^{\pm0.077}$ | $\mathbf{3.490}^{\pm0.061}$ |
| InterX | ComMDM (Shafir et al., 2024) | $0.090^{\pm0.002}$ | $0.165^{\pm0.004}$ | $0.236^{\pm0.004}$ | $29.266^{\pm0.067}$ | $6.870^{\pm0.017}$ | $4.734^{\pm0.067}$ | $0.771^{\pm0.053}$ |
| | InterGen (Liang et al., 2024) | $0.400^{\pm0.006}$ | $0.585^{\pm0.006}$ | $0.695^{\pm0.006}$ | $0.475^{\pm0.0305}$ | $3.800^{\pm0.020}$ | $9.095^{\pm0.055}$ | $2.657^{\pm0.090}$ |
| | InterMask (Javed et al., 2025) | $0.403^{\pm0.005}$ | $0.595^{\pm0.004}$ | $0.705^{\pm0.005}$ | $0.399^{\pm0.013}$ | $\underline{3.705}^{\pm0.017}$ | $9.046^{\pm0.073}$ | $2.261^{\pm0.081}$ |
| | TIMotion (Wang et al., 2025b) | $\underline{0.412}^{\pm0.004}$ | $\underline{0.601}^{\pm0.004}$ | $\underline{0.714}^{\pm0.003}$ | $\underline{0.385}^{\pm0.0218}$ | $3.706^{\pm0.015}$ | $\mathbf{9.191}^{\pm0.092}$ | $2.437^{\pm0.069}$ |
| | Ours | $\mathbf{0.416}^{\pm0.005}$ | $\mathbf{0.615}^{\pm0.004}$ | $\mathbf{0.718}^{\pm0.003}$ | $\mathbf{0.372}^{\pm0.025}$ | $\mathbf{3.704}^{\pm0.016}$ | $\underline{9.123}^{\pm0.075}$ | $\underline{2.978}^{\pm0.080}$ |

Table 1: Quantitative comparison of interaction-aware motion generation on the InterHuman and InterX datasets. We report Top-1/2/3 R-Precision (higher is better), FID and MM Distance (lower is better), Diversity, and MultiModality (higher is better). Our method consistently achieves competitive or superior performance across both datasets, demonstrating improvements in semantic alignment, motion realism, and interaction diversity. Best results among methods are highlighted in **bold**, and second-best results are underlined. The uncertainty (after ±) is shown as superscript.

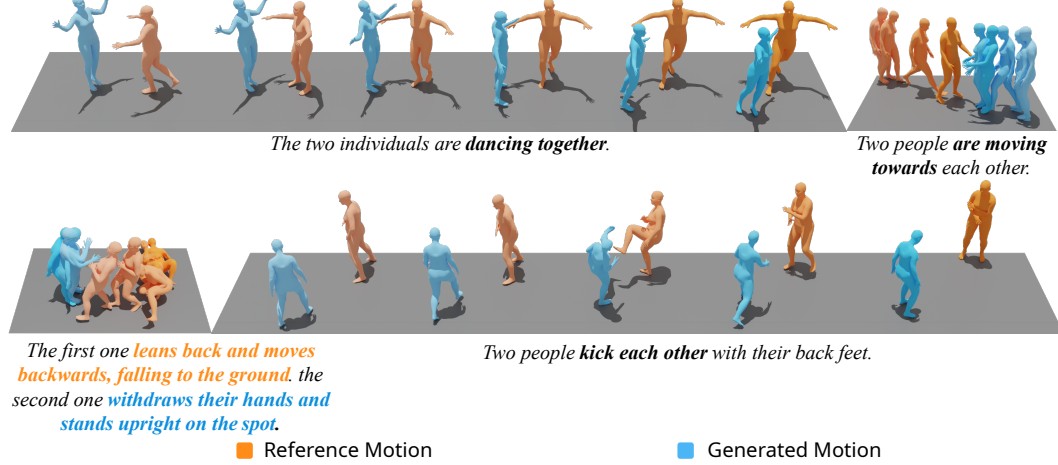

*The two individuals are **dancing together**.*

*Two people **are moving towards** each other.*

*The first one **leans back and moves backwards, falling to the ground**. the second one **withdraws their hands and stands upright on the spot**.*

*Two people **kick each other** with their back feet.*

🟧 Reference Motion   🟦 Generated Motion

Figure 3: Quantative visualization of generated reactions.

**Qualitative Comparison.** As illustrated in Figure 3, SyMoFlow produces temporally coherent and interaction-aware reactions. The model anticipates the first agent's actions and generates appropriate responses for the second agent, preserving natural joint trajectories and capturing interaction cues such as synchronized gestures, coordinated hand movements, or mutual avoidance. Compared to existing methods, our outputs exhibit more expressive and semantically consistent reactions, emphasizing interaction plausibility over mere physical consistency.

## 4.5 ABLATION STUDY

We conduct an ablation study on the InterHuman dataset to disentangle the contribution of the key components in SyMoFlow (Table 3). Compared with joint distribution modeling, a naive distribution decomposition that ignores inter-agent symmetry leads to degraded performance in both language-motion alignment (R-Precision) and motion fidelity (FID). This validates that our symmetric flow path formulation provides a principled way to preserve reciprocal dependencies between actors and

| Method | R-Precision↑ | | | FID↓ | MMDist↓ | Diversity→ | MModality↑ |
|---|---|---|---|---|---|---|---|
| | Top-1 | Top-2 | Top-3 | | | | |
| Ground Truth | $0.452^{\pm 0.008}$ | $0.610^{\pm 0.009}$ | $0.701^{\pm 0.008}$ | $0.273^{\pm 0.007}$ | $3.755^{\pm 0.008}$ | $7.948^{\pm 0.064}$ | – |
| InterMask (Javed et al., 2025) | $0.460^{\pm 0.005}$ | $0.620^{\pm 0.005}$ | $0.706^{\pm 0.006}$ | $3.026^{\pm 0.034}$ | $3.781^{\pm 0.001}$ | $7.760^{\pm 0.030}$ | $1.305^{\pm 0.041}$ |
| Ours | $\mathbf{0.475}^{\pm 0.007}$ | $\mathbf{0.636}^{\pm 0.006}$ | $\mathbf{0.711}^{\pm 0.005}$ | $\mathbf{2.782}^{\pm 0.044}$ | $\mathbf{3.772}^{\pm 0.001}$ | $\mathbf{7.843}^{\pm 0.028}$ | $\mathbf{2.627}^{\pm 0.087}$ |

Table 2: Quantitative comparison of reaction generation on the InterHuman dataset. Our method outperforms InterMask across most metrics, achieving higher R-Precision, Diversity, and Multi-Modality. While FID in Table 1 is slightly higher than InterMask for full sequence generation, in reaction generation FID is lower, reflecting more accurate modeling of interactions and improved motion realism. Best results are highlighted in **bold**.

| Method | R-Precision↑ | | | FID↓ | MMDist↓ | Diversity→ | MModality↑ |
|---|---|---|---|---|---|---|---|
| | Top-1 | Top-2 | Top-3 | | | | |
| Ground Truth | $0.452^{\pm 0.008}$ | $0.610^{\pm 0.009}$ | $0.701^{\pm 0.008}$ | $0.273^{\pm 0.007}$ | $3.755^{\pm 0.008}$ | $7.948^{\pm 0.064}$ | – |
| Joint Distribution Modeling | $\underline{0.486}^{\pm 0.005}$ | $\underline{0.639}^{\pm 0.005}$ | $\underline{0.718}^{\pm 0.004}$ | $6.249^{\pm 0.079}$ | $3.780^{\pm 0.001}$ | $\underline{7.967}^{\pm 0.039}$ | $\mathbf{2.930}^{\pm 0.097}$ |
| Naive Distribution Decomposition | $0.481^{\pm 0.005}$ | $0.634^{\pm 0.005}$ | $0.712^{\pm 0.005}$ | $6.719^{\pm 0.118}$ | $3.780^{\pm 0.001}$ | $\mathbf{7.946}^{\pm 0.035}$ | $2.771^{\pm 0.104}$ |
| Symmetric Flow Path | $0.482^{\pm 0.006}$ | $0.636^{\pm 0.005}$ | $0.716^{\pm 0.005}$ | $\underline{5.838}^{\pm 0.078}$ | $3.779^{\pm 0.001}$ | $\underline{7.953}^{\pm 0.037}$ | $2.791^{\pm 0.095}$ |
| SyMoFlow (Symmetric Flow Path + MCC) | $\mathbf{0.491}^{\pm 0.006}$ | $\mathbf{0.653}^{\pm 0.005}$ | $\mathbf{0.730}^{\pm 0.005}$ | $\mathbf{5.416}^{\pm 0.078}$ | $3.775^{\pm 0.001}$ | $\underline{7.953}^{\pm 0.037}$ | $\underline{2.844}^{\pm 0.095}$ |

Table 3: Ablation study on the InterHuman dataset, evaluating the contribution of key components in SyMoFlow. The table compares different design variants: joint distribution modeling, naive decomposition, symmetric flow path, and symmetric flow path with Motion Consistency Constraint (MCC). Best results are highlighted in **bold** and second-best results are underlined.

reactors, thereby capturing interaction-aware dynamics more faithfully. Further, while the symmetric flow path alone already improves fidelity over naive decomposition, it remains limited in enforcing temporal coherence across the two agents. Incorporating the Motion Consistency Constraint (MCC) effectively addresses this issue by explicitly regularizing causal agreement between action and reaction, leading to gains in both R-Precision and FID. Overall, the complete SyMoFlow framework achieves the best trade-off across all metrics, confirming that the integration of symmetry-aware flow design and consistency regularization is crucial for generating realistic, semantically aligned, and diverse human-human interactions.

## 5 CONCLUSION

We presented SyMoFlow, a task-decomposed framework for human-human interaction generation that jointly models motion and reaction using discrete flow matching on VQ-VAE latent codes. By decomposing the generation process into motion priors and reaction-conditioned distributions, our approach effectively captures inter-agent dependencies while maintaining diverse and realistic sequences. The introduction of the motion consistency constraint further enforces coherent and physically plausible interactions that adhere to textual prompts. Extensive experiments on Inter-Human and InterX demonstrate that SyMoFlow achieves competitive performance across fidelity (FID), semantic alignment (R-Precision, MM-Dist), and diversity/multimodality. Ablation studies validate the necessity of symmetry-aware decomposition and motion consistency, while qualitative evaluations highlight the model's ability to produce natural, contextually appropriate reactions.

**Limitations and Future Work.** SyMoFlow currently focuses on dyadic interactions with limited temporal scope. Future work includes extending the framework to multi-agent and longer-horizon scenarios, integrating stronger interaction and physical constraints to reduce artifacts such as inter-penetration, and exploring adaptive latent representations to further enhance diversity while maintaining realism.

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

## A  VQ-VAE TRAINING AND GEOMETRIC LOSSES

The Motion VQ-VAE is trained to encode and reconstruct motion sequences into a discrete latent space while enforcing physical plausibility. The overall training objective combines standard VQ-VAE losses with geometric regularization:

$$\mathcal{L}_{\text{VQ-VAE}} = \mathcal{L}_{\text{vq}} + \lambda_{\text{vel}}\mathcal{L}_{\text{vel}} + \lambda_{\text{fc}}\mathcal{L}_{\text{fc}} + \lambda_{\text{bl}}\mathcal{L}_{\text{bl}}, \tag{11}$$

where $\mathcal{L}_{\text{vq}}$ consists of reconstruction and commitment losses, and $\mathcal{L}_{\text{vel}}$, $\mathcal{L}_{\text{fc}}$, $\mathcal{L}_{\text{bl}}$ impose geometric and kinematic constraints.

Specifically, the geometric losses are defined as follows:

- **Velocity loss** $\mathcal{L}_{\text{vel}}$: Encourages reconstructed motions to match ground-truth joint velocities, improving temporal smoothness.

- **Foot contact loss** $\mathcal{L}_{\text{fc}}$: Encourages zero velocity for feet in contact with the ground, reducing sliding artifacts.

- **Bone length loss** $\mathcal{L}_{\text{bl}}$: Maintains consistency of bone lengths between adjacent joints to preserve anatomical plausibility.

Formally, given ground-truth poses $\mathbf{x}_n$ and reconstructed poses $\hat{\mathbf{x}}_n$ at time step $n$, these losses are computed as:

$$\mathcal{L}_{\text{vel}} = \frac{1}{N-1}\sum_{n=1}^{N-1}\left\|(\mathbf{x}_{n+1} - \mathbf{x}_n) - (\hat{\mathbf{x}}_{n+1} - \hat{\mathbf{x}}_n)\right\|_1,$$

$$\mathcal{L}_{\text{fc}} = \frac{1}{N-1}\sum_{n=1}^{N-1}\left\|(\hat{\mathbf{x}}_{n+1} - \hat{\mathbf{x}}_n) \cdot \mathbf{f}_n\right\|_1, \tag{12}$$

$$\mathcal{L}_{\text{bl}} = \frac{1}{N-1}\sum_{n=1}^{N-1}\left\|B(\mathbf{x}_n) - B(\hat{\mathbf{x}}_n)\right\|_1,$$

where $N$ is the sequence length, $\mathbf{f}_n \in \{0,1\}$ indicates foot contact for heel/toe joints, and $B(\cdot)$ computes bone lengths between adjacent joints. The coefficients $\lambda_{\text{vel}}$, $\lambda_{\text{fc}}$, and $\lambda_{\text{bl}}$ control the relative importance of each geometric regularizer.

This unified formulation ensures that the VQ-VAE not only reconstructs motion accurately, but also produces temporally smooth, physically plausible, and anatomically consistent sequences suitable for downstream discrete flow-based interaction generation.

## B  DISCRETE FLOW MATCHING DETAILS

### B.1  PRELIMINARIES

We build our motion generator upon the framework of Discrete Flow Matching (DFM) (Gat et al., 2024b). Formally, the goal of DFM is to transform a known source distribution $p(z)$ into a target data distribution $q(z)$ defined over a finite discrete space $S = \mathcal{T}^D$, where $D$ is the dimensionality of the discrete variable and $\mathcal{T} = [K] = \{1, 2, \ldots, K\}$ is the vocabulary of possible codebook indices from the VQ-VAE. Here, $z$ corresponds to motion tokens.

**Probability Paths**  Given the source distribution $p(z)$ and target distribution $q(z)$, DFM introduces a time-dependent probability path $\{p_t(z)\}_{t \in [0,1]}$ that smoothly interpolates between them:

$$p_t(z) = \sum_{z_1 \in S} p_t(z \mid z_1)q(z_1), \tag{13}$$

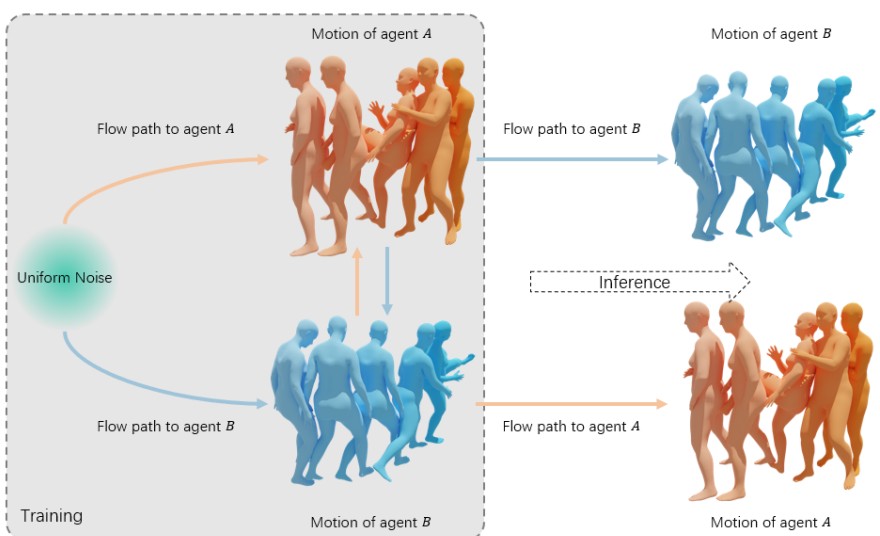

Figure 4: Discrete flow matching sampling in our framework. (a) During training, motion and reaction flows are coupled via a interaction-symmetric path, enabling bidirectional supervision between the two agents. (b) During inference, the process is sequential: a motion sequence for the first agent is generated from noise, and then the reaction of the second agent is generated conditioned on both the text prompt and the first agent's motion.

where the conditional distribution factorizes across dimensions,

$$p_t(z \mid z_1) = \prod_{i=1}^{D} p_t(z_i \mid z_{1,i}). \tag{14}$$

Each univariate interpolation $p_t(z_i \mid z_{1,i})$ is defined as a mixture path:

$$p_t(z_i \mid z_{1,i}) = (1 - \kappa_t(z_{1,i}))p(z_i) + \kappa_t(z_{1,i})\delta_{z_{1,i}}(z_i), \tag{15}$$

where $\kappa_t(\cdot) \in [0, 1]$ is a scheduler with $\kappa_0 = 0$ and $\kappa_1 = 1$, and $\delta_{z_{1,i}}(z_i)$ is a point mass at $z_{1,i}$. This design recovers the masked data construction when $p(z_i)$ is set to a mask token distribution.

**Probability Velocities**  To simulate the generative process along $\{p_t(z)\}_{t \in [0,1]}$, we consider a continuous-time Markov chain (CTMC) $\{z_t\}_{t \in [0,1]}$ over the discrete space, such that $z_t \sim p_t$. Its dynamics are governed by a probability velocity $u_t^i(\cdot, z_t)$ (rate matrix), describing the instantaneous transition rate of token $z_t^i$:

$$\sum_{z_i \in [K]} u_t^i(z_i, z_t) = 0, \quad u_t^i(z_i, z_t) \geq 0 \quad \forall i \in [D], \ z_i \neq z_{t,i}, \tag{16}$$

ensuring valid probability transitions. Moreover, $u_t$ satisfies the discrete continuity equation:

$$\frac{d}{dt}p_t(z) + \text{div}_z(p_t u_t) = 0, \tag{17}$$

which guarantees that the probability velocity field $u_t$ generates the prescribed probability path.

### B.2 SAMPLING PROCEDURE

During inference, each discrete token $z_t^i$ of agent $i$ at time $t$ is updated sequentially along the DFM path using an Euler-style solver:

1. Sample $z_1^i \sim p_{1|t}^i(\cdot \mid z_t^i)$ from the model as the target for the current step.

2. Compute the total conditional transition rate

$$\lambda^i = \sum_{z^i \neq z_t^i} u_t^i(z^i, z_t^i \mid z_1^i), \quad u_t^i(z^i, z_t^i | z_1^i) = \frac{\dot{\kappa}}{1 - \kappa}\Big[ p_{1|t}(z^i|z_t^i) - \delta_{z_t^i}(z^i) \Big], \qquad (18)$$

representing the intensity of probability mass flowing to other states.

3. Draw a uniform random variable $r_{\text{change}}^i \sim \mathcal{U}[0, 1]$.

4. Update $z_{t+h}^i$ as

$$z_{t+h}^i = \begin{cases} \text{sample from } \frac{u_t^i\big(\cdot, z_t^i|z_1^i\big)}{\lambda^i}(1 - \delta_{z^i}(\cdot)), & r_{\text{change}}^i \leq 1 - e^{-h\lambda^i}, \\ z_t^i, & \text{otherwise.} \end{cases}$$

If a jump occurs, the new token is drawn proportionally to $u_t^i(\cdot, z_t^i \mid z_1^i)$, biasing the update towards the model-predicted $z_1^i$.

This procedure allows continuous refinement of predictions along the probability path and repeated updates to tokens, improving interaction-aware reaction generation compared to mask-based discrete diffusion models.

## C   DETAILED TRANSFORMER BLOCK DESIGN

**Input Embeddings.**   Let $\mathbf{e}^{l-1} \in \mathbb{R}^{2nj \times \tilde{d}}$ denote the input token embeddings, including both persons' motion tokens and optionally a special token for text or time-step information.

**Global Self-Attention.**   We first apply multi-head self-attention over all tokens:

$$\text{Attn}(Q, K, V) = \text{Softmax}\Big(\frac{QK^\top}{\sqrt{\tilde{d}}}\Big)V, \qquad (19)$$

where $Q, K, V$ are linear projections of $\mathbf{e}^{l-1}$. The output is followed by a feed-forward network (FFN).

**Intra-Person Spatio-Temporal Attention.**   For each person $p \in \{a, b\}$, let $\mathbf{e}_p^{l-1} \in \mathbb{R}^{n \times j \times \tilde{d}}$ denote the embeddings for $n$ temporal steps and $j$ joints. We apply separate spatial and temporal attention:

- **Spatial attention:** for each time step $t$,
$$\mathbf{e}_p^{\text{spatial}}[t] = \text{Attn}\big(Q_p[t], K_p[t], V_p[t]\big), \qquad (20)$$
where $Q_p[t], K_p[t], V_p[t]$ are linear projections of the joint embeddings at step $t$, allowing each joint to attend to other joints in the same frame.
- **Temporal attention:** for each joint $j$,
$$\mathbf{e}_p^{\text{temporal}}[j] = \text{Attn}\big(Q_p[:, j], K_p[:, j], V_p[:, j]\big), \qquad (21)$$
where $Q_p[:, j], K_p[:, j], V_p[:, j]$ are linear projections along the temporal dimension, allowing each joint to attend to its own history across frames.

The outputs of spatial and temporal attention are summed to obtain the updated embeddings for person $p$:

$$\mathbf{e}_p' = \mathbf{e}_p^{\text{spatial}} + \mathbf{e}_p^{\text{temporal}}. \qquad (22)$$

**Inter-Person Cross-Attention.**   To model interactions with another agent, we apply cross-attention only for the generated agent (e.g., agent B):

$$\mathbf{e}_B' = \text{Attn}(Q_B, K_A, V_A), \qquad (23)$$

where $Q_B$ are linear projections of the updated embeddings $\mathbf{e}_B'$ after intra-person attention, and $K_A, V_A$ are projections of the conditioning agent's embeddings $\mathbf{e}_A$ (e.g., previously generated agent A motion).

The output $\mathbf{e}_B'$ is then passed through a feed-forward network (FFN) to obtain the final block output $\mathbf{e}_B^l$, which continues through subsequent transformer blocks.

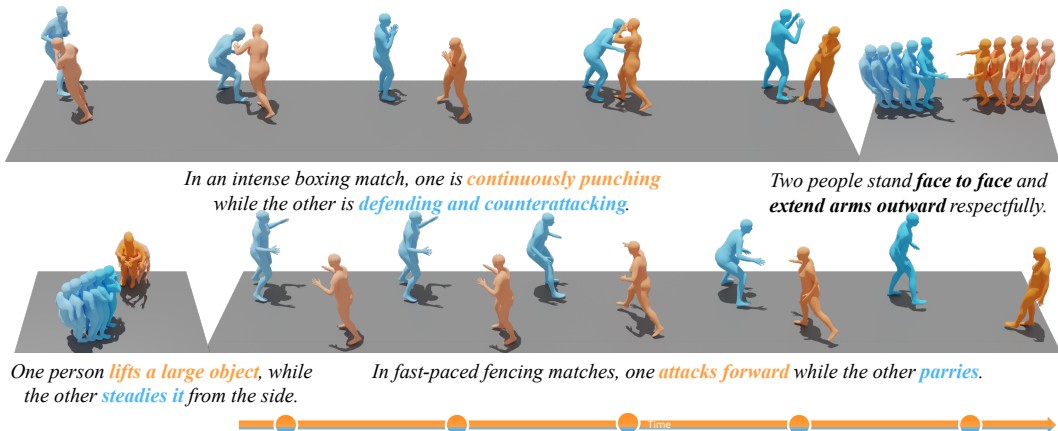

*In an intense boxing match, one is **continuously punching** while the other is **defending and counterattacking**.*

*Two people stand **face to face** and **extend arms outward** respectfully.*

*One person **lifts a large object**, while the other **steadies it** from the side.*

*In fast-paced fencing matches, one **attacks forward** while the other **parries**.*

Figure 5: Quantative visualization of generated human-human interactions.

**Auxiliary Motion Condition Injection.** The auxiliary condition $\mathbf{z}_{\text{cond}}$ is incorporated in two ways:

1. Concatenated to the token sequence in global self-attention.
2. Injected via cross-attention keys and values for inter-person attention.

**Text and Time-Step Conditioning.** Each attention and FFN module uses adaptive layer normalization (Peebles & Xie, 2023), where scale and shift parameters are regressed from the text embedding $c$ and time-step embedding.

**Output.** The updated embeddings for both individuals are passed through another FFN and concatenated to form the block output $\mathbf{e}^l \in \mathbb{R}^{2nj \times \tilde{d}}$, which is then passed to the next transformer block.

## D DATASETS

**InterHuman.** InterHuman follows the AMASS (Mahmood et al., 2019) skeleton representation with 22 joints, including the root. Each joint is described by global position $\mathbf{p}_g \in \mathbb{R}^3$, global velocity $\mathbf{v}_g \in \mathbb{R}^3$, and local 6D rotation $\mathbf{r}_{6d} \in \mathbb{R}^6$, yielding representations $\mathbf{x}_p \in \mathbb{R}^{N \times 22 \times 12}$.

**InterX.** InterX adopts the SMPL-X (Pavlakos et al., 2019) skeleton representation, comprising 54 body, hands, and face joints with root orientation and translation. Each joint and root orientation is represented by $\mathbf{r}_{6d}$, and root translation is described by $\mathbf{p}_g$ with velocity $\mathbf{v}_g$, resulting in $\mathbf{x}_p \in \mathbb{R}^{N \times 56 \times 6}$.

**Compatibility.** We adhere to the native body representations of both datasets, demonstrating that SyMoFlow is compatible with different skeleton formats and joint counts. Additional implementation details, including architecture, training, and inference hyper-parameters, are provided in Appendix E.

## E IMPLEMENTATION DETAILS

### E.1 MODEL ARCHITECTURE

Our models are implemented in PyTorch. The Motion VQ-VAE uses 2D convolutional residual blocks for both encoder and decoder. Temporal downsampling is set to $n/N = 1/4$ for both datasets, while spatial downsampling is dataset-specific: $j/J = 5/22$ for InterHuman and $j/J = 5/56$ for InterX. Strided convolutions perform downsampling in the encoder, and the decoder restores dimensions via upsampling followed by convolutional layers. The latent dimension is $d' = 512$ with a codebook size $|\mathcal{C}| = 1024$.

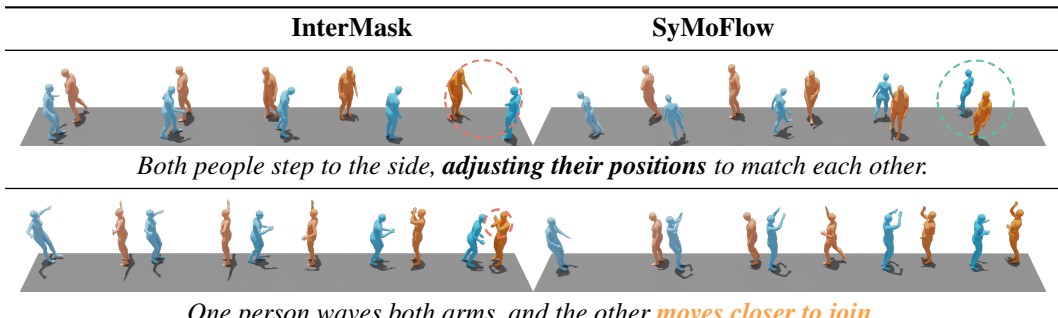

| InterMask | SyMoFlow |
|---|---|

*Both people step to the side, **adjusting their positions** to match each other.*

*One person waves both arms, and the other moves closer to join.*

Figure 6: More qualitative comparison between SyMoFlow and InterMask (Javed et al., 2025).

The Inter-M transformer uses $L = 6$ transformer blocks with 6 attention heads each, and the transformer embedding dimension is $\tilde{d} = 384$. Text conditioning is applied using CLIP ViT-L/14@336px features.

| Parameter | Value |
|---|---|
| $d'$ | 512 |
| $|\mathcal{C}|$ | 1024 |
| $n/N$ | 1/4 |
| $j/J$ (InterHuman) | 5/22 |
| $j/J$ (InterX) | 5/56 |
| $L$ | 6 |
| Attention heads | 6 |
| $\tilde{d}$ | 384 |
| CLIP version | ViT-L/14@336px |

Table 4: Model parameters for Motion VQ-VAE and Inter-M transformer.

### E.2 TRAINING DETAILS

The Motion VQ-VAE is trained following the general setting of InterMask (Javed et al., 2025), with modifications to better suit our datasets. Training runs for 50 epochs with a batch size of 512. The learning rate is initialized at 0.0002 and decayed by a factor of 0.1 at 70% and 85% of iterations, with a linear warm-up during the first 25% of iterations. The commitment loss factor $\beta$ is set to 0.02, and geometric losses for velocity, foot contact, and bone length are adjusted per dataset.

The flow field transformer largely follows the InterMask configuration but incorporates adjustments for our sequential interaction modeling. It is trained for 500 epochs with a batch size of 52, using a multi-step learning rate decay with factors of 1/3 at 50%, 70%, and 85% of iterations. A condition drop probability of 0.1 is applied to allow flexible training with or without text conditioning.

| Parameter | Value |
|---|---|
| VQ-VAE batch size | 512 |
| Transformer batch size | 36 |
| Initial learning rate | 0.0002 |
| Learning rate decay | 0.1 / 1 / 3 |
| $\beta$ | 0.02 |
| $\lambda_{\text{vel}}, \lambda_{\text{fc}}, \lambda_{\text{bl}}$ (InterHuman) | 100, 500, 5 |
| $\lambda_{\text{vel}}, \lambda_{\text{fc}}, \lambda_{\text{bl}}$ (InterX) | 100, 100, 5 |
| Condition drop prob. | 0.1 |

Table 5: Training hyperparameters for Motion VQ-VAE and Inter-M transformer.

### E.3    INFERENCE DETAILS

During inference, we report the number of function evaluations (NFE) instead of iterations. Since motion and reaction sequences are generated sequentially, the effective NFE is doubled. We use the same NFE, classifier-free guidance (CFG) scale, and temperature for both interaction and reaction generation: $n_{\text{fe}} = 20$, CFG scale = 2, and temperature = 1.

| Parameter | Value |
| --- | --- |
| NFE (effective) | $20 \times 2$ |
| CFG scale | 2 |
| Temperature | 1 |

Table 6: Inference hyperparameters. NFE is doubled to account for sequential generation of motion and reaction. The same settings are applied for both interaction and reaction sequences.

## F    VISUALIZATION RESULTS

More qualitative results are shown in Figure 6 and Figure 5. In addition, the supplementary materials include an extensive collection of video visualizations in the `videos/` directory, illustrating a broad spectrum of scenarios such as general interaction cases, close-proximity behaviors, reaction-based interactions, diverse generations conditioned on a single text prompt, multi-agent motion sequences, and representative failure cases. These videos provide a comprehensive view of the model's temporal coherence, semantic alignment, diversity, and multi-agent coordination capabilities.

