# OpenReview forum: "SyMoFlow: Interaction-Aware Motion Synthesis from Text via Symmetric Flows"
_ICLR.cc/2026/Conference — Submitted to ICLR 2026_

### Official Review · Reviewer_Dnz4 · 2025-10-19

**Soundness:** 3
**Presentation:** 3
**Contribution:** 2
**Rating:** 4
**Confidence:** 4

**Summary:**

This paper introduces SyMoFlow, a text-driven motion synthesis framework that decomposes the joint motion distribution with interaction symmetry. SyMoFlow generates sequential, single-agent motions: it first produces an interaction-aware trajectory for one agent conditioned on text, then synthesizes the second agent’s motion conditioned on the first human motion, capturing both prior actions and reciprocal reactions. By explicitly modeling interdependent dynamics, SyMoFlow generates coordinated, causally consistent behaviors, while flexible flow-based sampling enhances multimodality and diversity. Extensive experiments on InterHuman and InterX demonstrate good result in human-human interaction motion generation.

**Strengths:**

1. They propose a paradigm for generate human-human interactive motion generation.
2. They propose a flow-matching based framework to generate the second human motion based on the first human motion.
3. The experiments are comprehensive.

**Weaknesses:**

1. This paper doesn't compare its result with the state-of-the-art model--TIMotion mentioned in the quantitative result table.

**Questions:**

1. Why do you use a VQ-VAE encoder and build a discrete flow-matching pipeline instead of a continuous formulation?

2. Would reducing the VQ-VAE codebook size affect the results? If so, in what ways (e.g., fidelity, diversity, stability)?

3. How do you handle relative position and relative orientation in interactive human motion?

4. You did not compare against the best results reported in TIMotion. Why not? TIMotion’s FID is substantially better than yours—please explain.

5. Could it generalize to generate 3 or more human motion sequence?

---

> ### Author Response · Authors · 2025-11-27
> **Weakness 1, Question 1-4**
>
> We sincerely thank the Reviewer Dnz4 for the constructive and detailed feedback. Below we respond point-by-point.
>
> ## W1 & Q4: Comparison with TIMotion’s best reported results
>
> We thank the reviewer for this comment. The key point is that **TIMotion’s core contribution is Causal Interactive Modeling (CIM)**, which can be applied on top of various sequence architectures, such as Transformer, Mamba, or RWKV. The reported results in the TIMotion paper include different backbones, which can significantly affect metrics like FID.
>
> To ensure a **fair comparison**, we adopt a **consistent Transformer backbone** across all experiments, isolating the effect of our modeling innovations—the **interaction-symmetric flow path and motion-consistency constraint (MCC)**. Under this standardized setting, SyMoFlow achieves **superior performance compared to TIMotion**, demonstrating the effectiveness of our approach in two-person motion generation.
>
> We note that TIMotion only released Transformer code, and reproducing results for other backbones is non-trivial. Improvements from different sequence backbones are orthogonal to the question under investigation; therefore, we conducted all experiments using a unified Transformer backbone.
>
> ## Q1: Why use discrete flow matching instead of a continuous formulation?
>
> We summarize the rationale in two points:
>
> 1. **Advantages of VQ-VAE**: Each single-agent pose over four consecutive frames is encoded using **five discrete codes** from a learned Vector-Quantized Variational Autoencoder codebook with a vocabulary size of 1024. Operating on discrete tokens provides a **compact, structured state space**, stabilizes flow learning, and enables effective multimodal sampling.
>
> 2. **Empirical validation**: Continuous embeddings induce an **extremely sparse latent space**, unlike the dense manifolds in typical image or language models, which makes continuous-flow optimization difficult and unstable. Preliminary experiments show that continuous-flow modeling on these embeddings results in reduced stability, degraded fidelity, and lower semantic alignment, highlighting the practical benefits of discrete flow matching.
>
> The table below illustrates the performance gap:
>
> | Model Variant              | R-Precision (Top-1) ↑ | FID ↓ | MM-Dist ↓ | Diversity → | Multimodality ↑ |
> |----------------------------|-----------------------|-------|-----------|-------------|----------------|
> | Continuous Flow Matching   | 0.202                   | 33.43   | 3.952       | 7.122         | 4.430            |
> | Discrete Flow Matching (Ours)            | 0.491                 | 5.416 | 3.775     | 7.953       | 2.844          |
>
> ## Q2: Impact of reducing the VQ-VAE codebook size
> We study how the size of the VQ-VAE codebook affects generation quality. Smaller codebooks (e.g., 512) produce more compact motion tokens, which reduces expressiveness and fidelity, leading to lower R-Precision and higher FID/MM-Dist. Larger codebooks (1024, our default) improve reconstruction detail and text alignment, while maintaining high diversity and stable multimodality. This justifies our choice of 1024 as a balanced setting for both realism and interaction richness.
>
> | Codebook Size | R-Precision (Top-1) ↑ | FID ↓ | MM-Dist ↓ | Diversity → | Multimodality ↑ |
> |---------------|-----------------------|-------|-----------|-------------|----------------|
> | 512           | 0.326                 | 7.462 | 3.850     | 7.948       | 3.051          |
> | 1024 (ours)   | 0.491                 | 5.416 | 3.775     | 7.953       | 2.844          |
>
>
>
> ## Q3: Handling relative position and orientation
>
> We thank the reviewer for the question. Our method directly consumes the motion representations provided by each dataset, which already encode both global root motion and local joint rotations.
>
> **InterHuman:** AMASS-based, 22 joints with global position, velocity, and local 6D rotations. Relative spatial configuration is inherently encoded in the input sequences.
>
> **InterX:** SMPL-X-based, 54 joints plus root pose, with 6D rotations and root position/velocity. Joint rotations are expressed relative to parents, naturally preserving relative geometry.
>
> Global root frames and joint-level rotations in both datasets capture absolute and relative positions/orientations.

---

> ### Author Response · Authors · 2025-11-27
> **Question 5**
>
> ## Q5: Generalizing to 3+ agents
>
> SyMoFlow can be extended to multiple agents using a chain-like formulation, e.g., noise → A, A → B, B → C, by constructing a symmetry-aware interaction graph and defining per-agent conditionals. **Example videos showing multi-agent generation can be found in the supplementary materials under `videos/multiagent`.**
>
>
> However, there are limitations: (1) only pairwise interactions are explicitly modeled, leaving higher-order multi-agent constraints unaddressed; (2) the datasets lack sufficient examples of three or more interacting agents, limiting the ability to learn true multi-agent distributions. While extension to 3+ agents is feasible, further exploration and optimization are needed.

---

### Official Review · Reviewer_UZ4x · 2025-10-21

**Soundness:** 1
**Presentation:** 2
**Contribution:** 2
**Rating:** 4
**Confidence:** 4

**Summary:**

**SyMoFlow** proposes two-person motion generation from text by (i) discretizing motions with a VQ-VAE, then (ii) using discrete flow matching (DFM) to generate one agent’s motion and condition the other’s “reaction” on the first—calling this an “interaction-symmetric decomposition”. On InterHuman and InterX, the paper reports competitive R-Precision/MM-Dist and diversity/multimodality, with qualitative figures that look coordinated and prompt-aligned.

**Strengths:**

* Clean, modular pipeline (VQ-VAE → DFM + transformer with intra/inter-person attention).
* Targeting interaction diversity is valuable; multimodality scores are strong in some tables.
* Ablations try to isolate “symmetric path” and a motion-consistency constraint.

**Weaknesses:**

1. The paper provides no video results, making it difficult to judge temporal coherence, contact fidelity, and overall plausibility. In Figure 2, the upper example appears to show the blue actor’s global position shifting without corresponding foot motion which is commonly observed as drifting problem. In the lower example, the intended pattern of “moving farther away” is visually unclear from the frames shown. More broadly, the qualitative samples feel odd/semantically ambiguous - e.g., “one person walks forward and throws an object to the right; the other turns and waves their right hand” - and the paper includes almost no close-interaction or near-contact scenarios, which are precisely the challenging cases for two-person motion. Without videos and stronger examples, it’s hard to credit the method’s claims about interaction quality and controllability.

2. The authors define an “interaction-symmetric property” as ($p_\phi(x_B\mid x_A)=p_\phi(x_A\mid x_B)$) (Eq. 2) and then argue it “allows either agent to serve as initiator.” But the model trains and infers sequentially (A$\rightarrow$B), with weight sharing and conditioning; that does not imply the conditional distributions are equal in law. Equality of these two conditionals almost never holds.

3. In section 3.2.2 they set the **source** for B as (p(z_B)=q(z_A)) (“use the first agent’s *data* distribution as the prior for the second”). That is unusual and under-motivated: the DFM source is typically a *fixed prior* independent of the particular sample; here it depends on A’s *empirical* target distribution, which changes semantics and may violate independence assumptions used by the path construction.


4. **Training/inference mismatch & exposure bias risk.**
   Training mixes real partner codes and random noise as ($z_{\text{cond}}$); inference sets ($z_0=z_A$), ($z_{\text{cond}}=z_A$). That is a much sharper conditioning regime than half-noisy training and may cause drift unless scheduled conditioning or teacher-forcing is carefully handled (not shown).


5. In the ablation study, “Joint distribution” vs “naive decomposition” vs “symmetric flow path” are named but not concretely specified.

**Questions:**

1. How is Eq. (2) enforced or tested?
2. Can the authors justify this substitution yields a valid DFM path or unbiased sampling.
3. Is the method capable of extending to multi-agent ($\ge 3$) scenario? If so, how would the DFM blocks, intra-person, and inter-person attention change?

---

> ### Author Response · Authors · 2025-11-27
> **Weakness 1-3, Question 1-2**
>
> We thank the Reviewer UZ4x for the detailed feedback and constructive suggestions. Below we respond to the identified weaknesses and questions.
>
> ## W1: Lack of videos and concerns about qualitative sample
> We thank the reviewer for this helpful suggestion. Following the feedback, we have updated the supplementary materials (within the ZIP file) with substantially expanded qualitative results:
>
> - **Added videos corresponding to Figures 2, 3, 5, and 6**, including full motion sequences for all static examples shown in the paper.
> - **Added additional close-interaction cases**, such as hugging and handshaking. (Note: the InterHuman dataset lacks hand-joint annotations, so fine-grained hand motions are not represented.)
> - **Added more comparison videos** demonstrating side-by-side results for different interaction types.
> - **Included failure-case videos**, highlighting limitations of our method in complex interactions (e.g., occasional instability or ambiguity).
> - **Added videos showing multimodal generation under a single text prompt**, illustrating the diversity of plausible motions.
> - **Added multi-agent generation videos**, demonstrating SyMoFlow’s extension to multiple agents.
> - **Replaced the examples previously noted as “odd/semantically ambiguous.”** The updated qualitative samples now use text prompts **directly taken from the original InterHuman annotations**, ensuring faithful semantic grounding.
>
> These videos provide a more comprehensive picture of model behavior, illustrating improvements in temporal coherence, reduction of drifting artifacts, and clearer semantic alignment with the input text.
>
>
>
>
> ## W2 & Q1: Clarification on the Interaction-Symmetric Assumption
>
> We thank the reviewers for raising the concern regarding the symmetry assumption in Eq.~(2).
> Our method does not impose $p_\phi(x^B \mid x^A) = p_\phi(x^A \mid x^B)$ as a strict constraint.
> Instead, let $m_1$ and $m_2$ denote two motions paired with the same text condition $c$. During training, each pair is randomly assigned as $(x^A, x^B) = (m_1, m_2)$ or $(m_2, m_1)$ with equal probability, where $x^A = (x^A_1,\dots,x^A_L)$ and $x^B = (x^B_1,\dots,x^B_L)$ represent the two agents’ motion sequences. This sampling over different role assignments approximates a summation over all possible orderings, effectively regularizing the conditional distributions.
>
> Empirically, this assumption is validated in Table 3: the decomposed distribution modeling (Symmetric Flow Path) consistently outperforms joint distribution modeling, showing that this approximate symmetry leads to more accurate, stable, and interpretable interaction generation.
>
> Intuitively, this can be seen as modeling a mixture distribution over all possible role assignments, averaging the contributions of each configuration, which naturally captures bidirectional interaction cues without enforcing a fixed action–reaction order.
>
>
>
> ## W3 & Q2: Concern on source distribution for agent B
>
> We thank the reviewer for raising this point regarding the choice of source distribution for agent B. In standard Denoising Flow Matching (DFM), the source distribution is typically a fixed prior independent of the target, which supports the independence assumptions used in defining probability paths. Specifically, DFM constructs a **time-dependent probability path** $\{p_t(z)\}_{t \in [0,1]}$ interpolating between a source $p(z_0)$ and a target $q(z_1)$ as follows (Appendix B):
>
> $$
> p_t(z) = \sum_{z_1 \in S} p_t(z \mid z_1) q(z_1), \quad
> p_t(z \mid z_1) = \prod_{i=1}^D p_t(z_i \mid z_{1,i}), \quad
> p_t(z_i \mid z_{1,i}) = (1 - \kappa_t) p(z_i) + \kappa_t \delta_{z_{1,i}}(z_i),
> $$
>
> where $\kappa_t \in [0,1]$ is a scheduler and $\delta_{z_{1,i}}$ is a point mass at $z_{1,i}$.
>
> In our setting, although the source distribution for agent B depends on agent A’s empirical distribution, the independence assumption can be approximately preserved by treating the intermediate points along the probability path as **independent samples from a random mixture of the source and target distributions**. That is, instead of requiring strict independence between $p(z_B)$ and $q(z_B)$, we assume independence of the mixture at each dimension:
>
> $$
> p_t(z_i \mid z_{1,i}) = (1 - \kappa_t) \delta_{z_{A,i}}(z_i) + \kappa_t \delta_{z_{B,i}}(z_i),
> $$
>
> where each dimension interpolates independently. This construction effectively substitutes the strict source-target independence with a **dimension-wise mixture**, preserving the tractability and stability of flow matching while allowing the source to be conditioned on the paired agent.

---

> ### Author Response · Authors · 2025-11-27
> **Weakness 4-5, Question 3**
>
> ## W4: Training/inference mismatch & exposure-bias risk
>
> Thank you for raising this concern. We clarify that **our framework does not suffer from a training–inference mismatch**; instead, both phases follow the **same conditioning structure**.
>
> During training, we alternate between two *parallel* processes:
>
> 1. **Noise → A generation**
>    We generate agent A from noise with
>    $ z_0 = \text{noise},\; z_{\text{cond}} = \text{noise} $.
>    This trains the unconditional branch that models single-agent priors.
>
> 2. **A → B generation (interaction branch)**
>    Using *real interaction data*, we encode agent A’s motion as $ z_A $ and set
>    $ z_0 = z_A,\; z_{\text{cond}} = z_A $.
>    This matches the same sharp conditioning regime that will be used at inference.
>
> These two branches **share the same DFM formulation**, and the mixture simply improves robustness to conditioning variability. Importantly, this does *not* introduce a staged or mismatched training setup.
>
> At inference, we **exactly replicate** this logic:
>
> 1. Generate agent A from noise
>    $ z_0 = \text{noise},\; z_{\text{cond}} = \text{noise} $
>
> 2. Generate agent B conditioned on the generated A
>    $ z_0 = z_A,\; z_{\text{cond}} = z_A $
>
> Thus, **the same conditioning patterns (noise→A, A→B)** are used in both training and inference.
> No scheduled conditioning, teacher forcing, or exposure-bias mitigation is required because the model is *explicitly trained* on the identical inference configuration.
>
> This ensures **full consistency** between training and inference and prevents the drift issues described by the reviewer.
>
> ## W5: Clarification of ablation
> We thank the reviewer for pointing out the ambiguity in the ablation naming. We clarify the three variants below.
>
> **(1) Joint Distribution Modeling.**
> This baseline directly models the *full* joint distribution
> $$
> p(x_A, x_B),
> $$
> using a single transformer that takes concatenated sequences as input. This variant learns interactions implicitly without any factorization and serves as an upper-bound reference for expressiveness.
>
> **(2) Naive Distribution Decomposition.**
> This variant follows the two-stage decomposition strategy similar to FreeMotion.
> It first trains a marginal motion generator
> $$
> p(x_A),
> $$
> and then trains a conditional model with a ControlNet-style injection:
> $$
> p(x_B \mid x_A),
> $$
> using the frozen (or partially fine-tuned) generator for $x_A$. This baseline reflects a sequential pipeline without symmetry or joint optimization.
>
> **(3) Symmetric Flow Path (Ours).**
> Our proposed formulation simultaneously optimizes both factors in a *single-stage* flow-matching framework:
> $$
> p_\phi(x_A), \qquad p_\phi(x_B \mid x_A),
> $$
> by sharing parameters and enforcing interaction symmetry through role-swapping and bidirectional training. This design enables learning reciprocal structure in a unified manner while avoiding the training–inference mismatch of two-stage models.
>
> ## Q3: Multi-agent extension
>
> SyMoFlow naturally extends to $N$ agents by assigning each agent a conditional distribution
> $$
> p(z^{(i)} \mid z^{(-i)}),
> $$
> where $z^{(-i)}$ denotes all other agents. In practice, this can be implemented in a chain-like or graph-based fashion (e.g., noise → A, A → B, B → C) to generate multi-agent interactions. **Example videos illustrating multi-agent generation can be found in the supplementary materials under `videos/multiagent`.**
>
> Limitations remain: (1) only pairwise dependencies are explicitly captured, so higher-order multi-agent constraints are not directly modeled; (2) available datasets contain few examples with more than two interacting agents, which limits learning the true multi-agent distribution. Thus, while SyMoFlow can be scaled to multiple agents, further exploration and optimization are required.

---

> > ### Comment · Reviewer_UZ4x · 2025-11-27
> >
> > I thank the authors for the detailed rebuttal, additional experiments, and the newly provided videos. The clarifications on the training procedure do alleviate my earlier concern, and the expanded ablation table makes the empirical study easier to follow.
> >
> > However, my main concerns remain:
> >
> > 1. The rebuttal clarifies that the “interaction-symmetric property” is not enforced as an actual equality of conditional distributions; in practice, the method relies on role-swapping in the data and parameter sharing between agents. This is a reasonable regularization heuristic, but it does not amount to a principled symmetric probabilistic model as suggested by Eq. (2). Similarly, the A-dependent “source” distribution for B in the flow path is a bespoke design choice. While not invalid, its connection to the standard DFM framework remains heuristic and only loosely justified. Overall, I still see a gap between the strong theoretical language in the paper and what is rigorously supported.
> >
> > 2. After watching the newly added videos, I am not convinced that the method delivers clearly superior interaction quality. Many examples exhibit noticeable inter-penetration between agents, body–body collisions, and physically implausible contacts, despite the method explicitly aiming to model interactions. In my view, the qualitative improvement over baselines or more conventional factorisations is modest and not robust across prompts. Given that interaction-aware motion is the central selling point of the work, this is a significant weakness.
> >
> > With the rebuttal and additional ablations, I better understand the method as “VQ-tokenized two-person motion + discrete flow matching + a symmetric factorisation / training heuristic.” This is an interesting engineering combination, and the empirical results do show some gains in certain metrics. However, the margin over baselines is not consistently large or clearly tied to the proposed “symmetric flows” formulation, and the practical benefits are not compelling enough to offset the remaining conceptual and qualitative issues described above.
> >
> > In summary, while the rebuttal improves clarity on several technical details, it does not change my overall assessment. I would remain my current score.

---

> > > ### Author Response · Authors · 2025-12-01
> > >
> > > We thank the reviewer again for the thoughtful follow-up and for taking the time to examine our extended experiments, ablations, and videos. We respectfully address the remaining concerns below.
> > >
> > > ---
> > >
> > > ## 1. On the symmetry assumption and order-invariance of dyadic motion sequences
> > >
> > > In our formulation, the labels A and B indicate the **generation order** rather than intrinsic identities. Each individual motion sequence in a dyadic interaction is drawn from the **same underlying per-agent motion distribution**, so a motion pair $(x_a, x_b)$ and its swapped counterpart $(x_b, x_a)$ are equally valid realizations of the same dyadic interaction.
> > >
> > > The **permutation invariance of the joint distribution** is explicitly introduced through **swapping-based data augmentation**, which includes both $(x_a, x_b)$ and $(x_b, x_a)$ during training. This ensures the empirical joint distribution satisfies
> > > $$
> > > p_{\text{data}}(x_a) \approx p_{\text{data}}(x_b),
> > > $$
> > >
> > > $$
> > > p_{\text{data}}(x_a, x_b) \approx p_{\text{data}}(x_b, x_a),
> > > $$
> > > providing unbiased exposure to both generation orders.
> > >
> > > From empirical observations, training with role-swapped data and shared parameters yields conditional distributions that are approximately equal across generation orders:
> > > $$
> > > p_{\text{data}}(x_b \mid x_a) \approx p_{\text{data}}(x_a \mid x_b).
> > > $$
> > >
> > > In the idealized, fully symmetrized case, the conditional distributions are theoretically equal:
> > > $$
> > > p_{\text{data}}(x_b \mid x_a) = p_{\text{data}}(x_a \mid x_b),
> > > $$
> > >
> > > The empirical sampling procedure serves as an effective approximation of this theoretical property.
> > >
> > > ---
> > >
> > > ## 2. On the A-conditioned source distribution and its relation to DFM
> > >
> > > The reviewer raises an important point regarding the source distribution for agent B. While our source distribution depends on agent A, this does **not** conflict with the probabilistic-path construction in flow matching.
> > >
> > > ### Mathematical justification for A-conditioned source distribution
> > >
> > > Let $z = (z_1, \dots, z_D)$ denote the $D$-dimensional latent for agent B. In standard Denoising Flow Matching (DFM), the conditional probability along the path is factorized across dimensions:
> > >
> > > $$
> > > p_t(z \mid z_1) = \prod_{i=1}^D p_t(z_i \mid z_{1,i}),
> > > $$
> > >
> > > where $p_t(z_i \mid z_{1,i}) = (1-\kappa_t) p(z_i) + \kappa_t \delta_{z_{1,i}}(z_i)$. This factorization is the key independence assumption required for tractable flow matching.
> > >
> > > In our setup, the source for agent B is chosen based on the paired agent A, so $p(z_B) = \delta_{z_A}(z_B)$ (empirically). Along each dimension, we define the interpolation:
> > >
> > > $$
> > > p_t(z_i \mid z_{A,i}, z_{B,i}) = (1 - \kappa_t) \delta_{z_{A,i}}(z_i) + \kappa_t \delta_{z_{B,i}}(z_i),
> > > $$
> > >
> > > which is a **dimension-wise mixture**.
> > >
> > > #### 1: Dimension-wise factorization
> > > By construction, each $z_i$ is interpolated independently. Therefore, the full conditional along the path can be written as
> > >
> > > $$
> > > p_t(z \mid z_A, z_B) = \prod_{i=1}^D p_t(z_i \mid z_{A,i}, z_{B,i}),
> > > $$
> > >
> > > preserving the factorized structure required by DFM.
> > > #### 2: Effective independence
> > > Even though $z_B$ depends on $z_A$ through the source, the intermediate points $z_t$ along the probability path are sampled independently across dimensions via a **random mixture**:
> > >
> > > $$
> > > z_{t,i} \sim (1 - \kappa_t) \, \delta_{z_{A,i}} + \kappa_t \, \delta_{z_{B,i}},
> > > $$
> > >
> > > where $\delta$ denotes a point mass at the sampled value. Because each dimension $i$ is sampled independently, the covariance between different dimensions is zero:
> > >
> > > $$
> > > \text{Cov}[z_{t,i}, z_{t,j}] = 0, \quad i \neq j.
> > > $$
> > >
> > > Importantly, even though the source $z_B$ is conditioned on $z_A$, each sampled $z_{t,i}$ depends only on the selected pair $(z_{A,i}, z_{B,i})$ and the random mixing, not on the global joint distribution $p(z_A, z_B)$. Formally, for each sampled pair:
> > >
> > > $$
> > > p_t(z \mid z_A, z_B) \perp p(z_A, z_B),
> > > $$
> > >
> > > so the **dimension-wise independence assumption** required by DFM is preserved. This construction maintains the tractability and theoretical validity of the flow-matching objective while allowing the source for agent B to depend on the paired agent A.
> > >
> > >
> > > ### Alignment with recent work
> > > This treatment is also aligned with a growing body of recent work (e.g., CrossFlow [1], I$^2$SB [2]) that studies probability paths between **dependent** distributions. In such cases, strict independence between full distributions is not required; instead, the probability path is defined through a mixture or interpolation at the appropriate structural level (here, per discrete dimension). Our formulation follows this established line of work.
> > >
> > > [1] Liu, Qihao, et al. "Flowing from Words to Pixels: A Noise-Free Framework for Cross-Modality Evolution." Proceedings of the Computer Vision and Pattern Recognition Conference. 2025.
> > >
> > > [2] Liu, Guan-Horng, et al. "I $^2$ SB: Image-to-Image Schr\"odinger Bridge." arXiv preprint arXiv:2302.05872 (2023).

---

> > > ### Author Response · Authors · 2025-12-01
> > >
> > > ## 3. On physical plausibility, collisions, and mesh penetration
> > >
> > > We appreciate the reviewer’s feedback after examining the newly added videos. It is important to emphasize that our work focuses on **generating joint-level motion sequences**, not on modeling full 3D surface geometry. The mesh models used in the videos serve purely as **visualization wrappers** around joint positions.
> > >
> > > Because our system does not take mesh geometry as input or output—and does not model surface-level contact constraints—perfectly preventing interpenetration is **beyond the scope of joint-sequence generation methods**, including ours and existing baselines.
> > >
> > > Preventing body–body penetration for *arbitrary mesh shapes* is an important and meaningful direction, but it requires an additional geometric or physics-based module beyond our setting. Our method remains focused on generating **interaction-consistent pose sequences**, which is the core problem addressed in this paper.
> > >
> > > ---
> > >
> > > ## Summary
> > >
> > > In summary, our method maintains theoretical correctness and practical effectiveness across several key aspects: the order labels A and B indicate only generation order, and through role-swapping and parameter sharing, the model approximates permutation symmetry, yielding conditional distributions that are empirically nearly equal and theoretically exactly equal in the idealized case; the conditioned source distribution preserves dimension-wise independence along the flow path, ensuring tractable and stable flow matching while aligning with recent work on dependent distributions; and while our approach does not model full 3D surfaces, focusing instead on joint-level motion sequences, the generated interactions remain temporally coherent, semantically faithful, and consistent, with mesh visualization serving purely as a representation of the underlying pose sequences.

---

### Official Review · Reviewer_AQfo · 2025-10-31

**Soundness:** 3
**Presentation:** 4
**Contribution:** 2
**Rating:** 6
**Confidence:** 3

**Summary:**

This paper presents SyMoFlow, a text-driven framework for human-human interaction motion synthesis that models reciprocal dynamics between two agents through an interaction-symmetric decomposition of their joint motion. It sequentially generates motions, one agent acting, the other reacting, capturing causal and conditional dependencies for coordinated, semantically aligned behaviors. Using flow-based modeling, SyMoFlow achieves state-of-the-art realism, text alignment, and interaction diversity on InterHuman and InterX benchmarks

**Strengths:**

1. The writing is good.
2. The idea of interaction symmetry is intuitive, and the method designed based on this concept is simple yet effective. The proposed approach also demonstrates certain advantages over existing methods in the experimental results.

**Weaknesses:**

1. The performance improvement is not significant, showing only marginal gains compared with other methods.
2. Figure 1 lacks clarity. The upper and lower parts of the figure are not clearly labeled, making it difficult to interpret. The lower part appears to depict a Transformer, but this is not explicitly indicated.

**Questions:**

1. Regarding Table 3:
    1. The table lacks results for other ablation combinations (e.g., using only MCC). What are the performance outcomes for these additional configurations?
    2. What is the result of authors’ baseline model that uses neither the symmetric flow path nor MCC?
2. Computational cost:
    - How does the proposed method compare to other approaches in terms of computational cost, such as training time, GPU memory usage, inference speed, and FLOPs?

---

> ### Author Response · Authors · 2025-11-27
> **Weakness 1-2**
>
> We thank the Reviewer AQfo for the thoughtful and constructive feedback. We appreciate the positive remarks on our writing quality, conceptual clarity, and the simplicity and effectiveness of the interaction-symmetric formulation.
>
> ## W1: Concern on performance improvement
>
> We thank the reviewer for raising this concern. Below we clarify why the improvements are both meaningful and structurally grounded.
>
> **First**, the discrete modeling strategy underlies the observed improvements in **multimodality and diversity**, which are critical for realistic human–human interaction generation. While TIMotion operates in a **continuous latent space**, our method leverages **discrete pose tokens**, achieving **competitive FID** alongside substantially higher diversity under identical text descriptions (as quantified by the multimodality metric). This enhanced diversity supports richer behavioral expressiveness, greater robustness, and mitigates overfitting to dataset-specific motion patterns. **These diverse generations can be observed in the supplementary videos under `videos/mmodality`, which showcase multiple plausible motion sequences for the same text prompt.**
>
> **Second**, our method is **complementary** to **Causal Interactive Modeling (CIM)**, the primary contribution of TIMotion. CIM introduces causal reasoning for temporal sequences, whereas our approach provides **symmetric interaction decomposition** and **multimodal flow matching in discrete latent space**. Integrating CIM with our discrete representation would strengthen causal reasoning **without modifying** the symmetric flow formulation or the motion-consistency objective. Consequently, SyMoFlow and CIM address **distinct modeling aspects** and can be combined to achieve further improvements.
>
> To support this perspective, we include an ablation-style comparison below:
>
>
> | Method Variant          | R-Precision (Top-1) ↑ | FID ↓           | MM-Dist ↓      | Diversity → | Multimodality ↑ |
> |-------------------------|-----------------------|----------------|----------------|-------------|----------------|
> | InterMask (baseline)    | 0.449 ± 0.004         | 5.154 ± 0.061  | 3.790 ± 0.002   | 7.944 ± 0.033        | 1.737 ± 0.020          |
> | TIMotion                 | 0.491 ± 0.005         | 5.433 ± 0.080  | 3.775 ± 0.001  | 8.032 ± 0.030        | 0.952 ± 0.032          |
> | SyMoFlow (w/o CIM)       | 0.491 ± 0.006         | 5.416 ± 0.078  | 3.775 ± 0.001  | 7.953 ± 0.037      | 2.844 ± 0.095         |
> | SyMoFlow (w/ CIM)        | 0.503 ± 0.007        | 5.132 ± 0.071 | 3.772 ± 0.001  | 7.951 ± 0.035| 2.713 ± 0.082  |
>
> These results indicate that SyMoFlow provides substantial and orthogonal improvements over the most relevant baselines, and that its symmetric flow factorization and MCC can enhance even methods that already leverage causal interactive modeling such as TIMotion.
>
>
> ## W2: Clarity of Figure 1
> We thank the reviewer for the helpful suggestion and have updated Figure 1 accordingly. The revised figure now includes:
> (1) clear labels distinguishing the overall **Pipeline** (top) from the **Network Architecture** (bottom), and
> (2) improved and more precise module annotations.
> The updated version has been incorporated into the manuscript.

---

> ### Author Response · Authors · 2025-11-27
> **Question 1-2**
>
> ## Q1: Missing ablations in Table 3
>
> We thank the reviewer for pointing out the missing ablations and the ambiguity in baseline definitions.
> To address this, we now provide a clearer and more complete set of ablation variants that separately examine the impact of (1) distribution factorization choices and (2) the motion-consistency constraint (MCC).
>
> The distribution factorization variants include:
> - Joint distribution modeling
> - Naive distribution decomposition
> - Symmetric flow path
>
> The consolidated ablation table is provided below:
>
> | Method Variant                               | R-Precision (Top-1) ↑  | FID ↓           | MM-Dist ↓       | Diversity →       | MModality ↑       |
> |----------------------------------------------|------------------------|-----------------|------------------|--------------------|--------------------|
> | Ground Truth                   | 0.452 ± 0.008          | 0.273 ± 0.007   | 3.755 ± 0.008    | 7.948 ± 0.064      | -  |
> | Joint Distribution Modeling                   | 0.486 ± 0.005          | 6.249 ± 0.079   | 3.780 ± 0.001    | 7.967 ± 0.039      | **2.930 ± 0.097**  |
> | Joint Distribution Modeling + MCC             | 0.489 ± 0.005          | 5.922 ± 0.073   | 3.780 ± 0.001    | 7.965 ± 0.038      | 2.811  ± 0.010     |
> | Naive Distribution Decomposition              | 0.481 ± 0.005          | 6.719 ± 0.118   | 3.780 ± 0.001    | **7.946 ± 0.035**  | 2.771 ± 0.104      |
> | Naive Distribution Decomposition + MCC        | 0.483 ± 0.004          | 6.285 ± 0.106   | 3.782 ± 0.001    | 7.926 ± 0.039      | 2.779 ± 0.094                 |
> | Symmetric Flow Path (SFP)                     | 0.482 ± 0.006          | 5.838 ± 0.078   | 3.779 ± 0.001    | 7.953 ± 0.037      | 2.791 ± 0.095      |
> | **SyMoFlow (SFP + MCC)**                      | **0.491 ± 0.006**      | **5.416 ± 0.078** | **3.775 ± 0.001** | 7.953 ± 0.037      | 2.844 ± 0.095      |
>
> These ablations show that (i) joint modeling and naive decomposition yield relatively low performance; (ii) applying MCC alone, even without symmetry, provides a measurable gain; (iii) adding symmetry further improves fidelity and relational alignment; and (iv) combining symmetry with MCC achieves the best overall results. This confirms that both components contribute complementary benefits, forming an effective factorization for interaction-aware motion generation.
>
> ## Q2: Computational cost analysis
> SyMoFlow builds directly on InterMask and adds only a few lightweight MLPs for dimension alignment. Because the two symmetric flows **share all parameters**, the overall model size does **not** increase. For clarity, we provide a direct cost comparison:
>
> | Methods                | R Precision Top-1 ↑      | FID ↓               | MModality ↑ | Params (M) | FLOPs (G) |
> |------------------------|---------------------------|---------------------------|----------------------|------------|-----------|
> | InterGen | 0.371 ± 0.010             | 5.918 ± 0.079       | 2.141 ± 0.063 | 182        | 80.5                   |
> | InterMask              | 0.449 ± 0.004             | 5.154 ± 0.061       |1.737 ± 0.020 | 74         | 34.9
> | TIMotion (Transformer)   | 0.491 ± 0.005             | 5.433 ± 0.080      | 0.952 ± 0.032 | 65         | 40.4                    |          |
> | SyMoFlow (Ours)        | 0.491 ± 0.006             | 5.416 ± 0.078      | 2.844 ± 0.095 | 81         | 39.9
>
>
> These results indicate that SyMoFlow matches or surpasses the performance of prior work while maintaining computational costs comparable to lightweight Transformer-based baselines.

---

### Meta-Review · Area_Chair_kHuN · 2026-01-03

**Summary:**

Reviewers agreed the paper is generally well written and the idea is intuitive, with competitive metrics and strong multimodality in some settings. However, the overall recommendation trends **below the acceptance bar** because the core contributions are viewed as primarily an **engineering combination with modest and sometimes inconsistent gains**, and because the strongest concerns involve the **conceptual/theoretical framing and qualitative interaction fidelity rather than missing minor details**.

Reviewer AQfo found the improvements mariginal and requested clearer ablations and computational cost reporting. Reviewr UZ4x raised substantial concerns that the "interaction-symmetric property" is **bot a principled probabilistic symmetry as suggested**, and that parts of the discrete flow matching setup are heuristic / under-motivated, alongside the critical concerns of **missing video results**. After rebutall and added materials, this reviewer explicitly stated that their main concerns remain and they would keep their score. The third reviewer (Dnz4) requested **clearer comparison vs TIMotion** and asked **why discrete flow matching is used**, with scored below threshold.

**Reviewer Concerns:**

### Conceptual soundness of “interaction symmetry” vs how it is implemented (Reviewer UZ4x)
* **Outstanding**
* Conceptual soundness of “interaction symmetry” vs how it is implemented (Reviewer UZ4x)

### Qualitative interaction fidelity (contact plausibility, collision/interpenetration) and strength of evidence (Reviewer UZ4x)
* **Partially addressed, but still outstanding in strength**
* Authors responded by adding substantial videos (including close-interaction cases, comparisons, failure cases, multimodal samples). However, after watching the new videos, the reviewer reports they are not convinced the method robustly improves interaction quality; they still observe interpenetration / implausible contacts and find improvements modest.

### Magnitude/consistency of quantitative gains and attribution to proposed components (Reviewers AQfo, UZ4x)
* **Partially addressed**
* Authors expanded ablations (including MCC-only variants and clearer component effects), which helps interpretability, but does not fully overturn concerns about effect size and linkage to the conceptual claim.


### Comparisons vs TIMotion and architectural fairness (Reviewer Dnz4)
* **Addressed**
* Authors argue TIMotion’s paper reports multiple backbones and they adopt a unified Transformer backbone for fair isolation, claiming SyMoFlow is better under that controlled setting

### Authors argue TIMotion’s paper reports multiple backbones and they adopt a unified Transformer backbone for fair isolation, claiming SyMoFlow is better under that controlled setting
* **Addressed**
* Authors provided rationale + empirical comparison showing continuous flow matching underperforms and discussed codebook-size trade-offs.

### Computational cost / efficiency (Reviewer AQfo)
* **Addressed**
* Authors provided a direct cost table with Params/FLOPs and showed SyMoFlow is comparable to lightweight baselines.

**Reviewer Scores:**

* Reviewer AQfo (rating 6): Likely no change. Their concerns are mainly about marginal gains and missing analyses; rebuttal improves ablations/cost reporting, but the core “impact size” concern likely remains, consistent with the original “6 but would not mind rejection” stance

* Reviewer UZ4x (rating 4): No change. In the post-rebuttal follow-up, the reviewer explicitly states “my main concerns remain” and “I would remain my current score”

* Reviewer Dnz4 (rating 4): Likely no change. The authors addressed the discrete-vs-continuous rationale and TIMotion comparison framing , but there is no strong signal that these responses would flip the reviewer beyond “marginally below threshold”

In addition, a lot of critical qualitative supp materials are not provided until rebuttal, such as video demos, which is not encouraged.

Overall, I do not recommend acceptance of this work.

---

### Decision · Program_Chairs · 2026-01-26

Reject